# Optical Gain of Vertically Coupled Cd_0.6_Zn_0.4_Te/ZnTe Quantum Dots

**DOI:** 10.3390/nano13040716

**Published:** 2023-02-13

**Authors:** Ming Mei, Minju Kim, Minwoo Kim, Inhong Kim, Hong Seok Lee, Robert A. Taylor, Kwangseuk Kyhm

**Affiliations:** 1Department of Optics & Cogno-Mechatronics Engineering, Pusan National University, Busan 46241, Republic of Korea; 2Smart Gym-Based Translational Research Center for Active Senior’s Healthcare, Pukyong National University, Busan 48513, Republic of Korea; 3Department of Physics, Jeonbuk National University, Jeonju 54896, Republic of Korea; 4Department of Physics, University of Oxford, Oxford OX1 3PU, UK

**Keywords:** optical modal gain, amplified spontaneous emission, CdZnTe, exciton, quantum dots

## Abstract

The optical modal gain of Cd_0.6_Zn_0.4_Te/ZnTe double quantum dots was measured using a variable stripe length method, where large and small quantum dots are separated with a ZnTe layer. With a large (~18 nm) separation layer thickness of ZnTe, two gain spectra were observed, which correspond to the confined exciton levels of the large and small quantum dots, respectively. With a small (~6 nm) separation layer thickness of ZnTe, a merged single gain spectrum was observed. This can be attributed to a coupled state between large and small quantum dots. Because the density of large quantum dots (4 × 10^10^ cm^−2^) is twice the density of small quantum dots (2 × 10^10^ cm^−2^), the density of the coupled quantum dots is determined by that of small quantum dots. As a result, we found that the peak gain (123.9 ± 9.2 cm^−1^) with the 6 nm separation layer is comparable to that (125.2 ± 29.2 cm^−1^) of the small quantum dots with the 18 nm separation layer.

## 1. Introduction

Cd_x_Zn_1 − x_Te is useful for various optoelectronic applications such as visible detectors (1.45~2.26 eV) [1,2], solar cells [3], X-ray, and gamma-ray detectors [4], where the band gap can be tailored by changing the concentration of Zn content. In particular, Cd_x_Zn_1 − x_Te quantum dots (QDs) have been extensively investigated; the interband transition energy was adjusted effectively by varying the growth conditions [5], and the QD size was controlled by changing Cd content as well as deposited layer thickness [6,7].

Recently, this technique enabled the growth of double quantum dot (DQD) structures [8,9], where the large quantum dots (LQDs) and small quantum dots (SQDs) of Cd_x_Zn_1 − x_Te are separated by a separation layer of ZnTe. When the separation layer thickness of ZnTe becomes large, the tunneling between LQDs and SQDs becomes suppressed. In this case, the SQDs and LQDs give rise to two separate emission spectra. On the other hand, as the separation layer thickness decreases (<10 nm), the tunneling is enhanced significantly. As a result, a single emission spectrum appears due to the coupling between LQDs and SQDs [7]. In addition, the strain distribution and interband transitions of Cd_x_Zn_1 − x_Te/ZnTe DQDs with different strength of coupling was also studied using a three-dimensional finite difference method (FDM) calculation, whereby strain and non-parabolicity effects were shown to play an important role in determining the electronic subband energies of Cd_x_Zn_1 − x_Te/ZnTe DQDs [10].

Although the presence of the coupled state in Cd_x_Zn_1 − x_Te DQDs was investigated mainly by photoluminescence (PL) under weak excitation, DQDs are also useful as a multi-color laser medium. In this case, amplified spontaneous emission (ASE) needs to be analyzed under strong excitation. However, gain analysis based on the ASE spectrum has rarely been studied, and is essential in order to evaluate the gain medium for laser diode applications. In this work, we used a variable stripe length method (VSLM) to investigate the optical modal gain of Cd_0.6_Zn_0.4_Te/ZnTe DQDs with two different ZnTe separation layers (18 nm and 6 nm). PL and ASE spectra at 5 K were measured, and the stripe length dependence of the ASE intensity was analyzed at various emission energies. This enables us to compare the two optical gain spectra with different separation layer thicknesses.

## 2. Materials and Methods

A schematic of two different sample structures containing Cd_0.6_Zn_0.4_Te/ZnTe DQDs is shown in Figure 1a. The samples were grown on GaAs substrates by molecular beam epitaxy (MBE), and consisted of the following structures; a ZnTe buffer layer (900 nm), a layer of Cd_0.6_Zn_0.4_Te SQDs, a ZnTe separation layer, a layer of Cd_0.6_Zn_0.4_Te LQDs, and a capping layer (100 nm). The diameters of the SQDs and LQDs were 40 ± 5 nm and 50 ± 5 nm, respectively. Because the height of the SQDs and LQDs (~10 nm) was similar, the difference in the confinement energy levels was determined mainly by the lateral size. When two QDs with different sizes are near each other, quantum tunneling between the different confinement energy levels needs to be considered. Given that the predicted efficient tunneling range is < 10 nm, we prepared two different separation distances (*d* = 6 nm, 18 nm) between SQDs and LQDs. With an 18 nm thickness for the ZnTe separation layer, we found that the tunneling was suppressed. On the other hand, with a 6 nm ZnTe separation layer, the two wavefunctions of the LQDs and SQDs can overlap, giving rise to energy levels of a coupled QD system.

In Figure 1b,c, PL spectra of Cd_0.6_Zn_0.4_Te DQDs with different separation distances (*d* = 6 nm, 18 nm) were measured at 5 K, respectively. With the large separation distance of *d* = 18 nm in Figure 1b, the PL spectra of the LQDs and SQDs are distinct, where the dominant PL peaks appear at 2.157 eV (LQDs) and 2.248 eV (SQDs), respectively.

When considering the confinement levels of the LQDs and SQDs, the diameter difference between 40 nm and 50 nm dots is expected to cause a few meV ground state differences. However, the peak-to-peak energy difference between the two PL spectra is 91 meV. Recently, it has been shown that valence band intermixing in CdZnTe by strain can also cause tens of meV of splitting [11,12,13]. When the ZnTe separation distance decreases, the strain applied to the QDs increases. As a result, the energy difference between the two confinement energy levels should also increase. Therefore, a peak-to-peak energy difference of 91 meV supports the presence of strain in our QDs.

As the separation layer thickness decreases, the tunneling efficiency increases significantly, resulting in a good wavefunction overlap. In this case, a single confinement energy level is seen for the coupled system, which appears in between the two different confinement energy levels of the LQDs and SQDs. In other words, the coupled system should be considered a single structure instead of two separate QDs. As shown in Figure 1c, a merged single PL spectrum appears near 2.187 eV when the separation thickness is 6 nm. This is evidence of the coupled energy level between LQDs and SQDs.

Figure 1d shows an experimental setup for optical gain. The sample was excited by a nanosecond pulsed laser (355 nm) with 4 mW excitation power. A cylindrical lens was used to prepare an optical stripe, and a movable beam block was used to change the stripe length (x). Along the beam stripe, the propagating emission is amplified, and ASE is emitted at the sample edge with a solid angle Ω. Therefore, the stripe length dependence of the ASE intensity, $I_{\mathrm{ASE}\;}\left(\hbar \omega ,x\right)$ can be described by [14]:(1)$$\frac{dI_{\mathrm{ASE}}\left(\hbar \omega ,\;x\right)}{dx}=g\left(\hbar \omega \right)I\left(\hbar \omega ,\;x\right)+J_{\mathrm{spon}}\left(\hbar \omega \right)\Omega ,$$
where g(ℏω) and $J_{\mathrm{spon}}$ are the modal gain coefficient at a selected emission energy (ℏω) and the spontaneous emission density, respectively. Supposing the optical gain coefficient is independent of stripe length, a solution of Equation (1) is then given by [15]:
(2)$$I_{\mathrm{ASE}}\left(\hbar \omega ,x\right)=\frac{J_{\mathrm{spon}}\left(\hbar \omega \right)\Omega }{g\left(\hbar \omega \right)}(e^{g\left(\hbar \omega \right)x}-1).$$

Provided that the ASE intensity $I_{\mathrm{ASE}}\left(\hbar \omega ,x\right)$ shows an exponential growth with increasing stripe length (x) for a selected emission energy (ℏω), the optical gain coefficient at a particular energy (ℏω) can be obtained by fitting the intensity with Equation (2). It is noticeable that g(ℏω) is a modal gain, where mode confinement (Γ) and optical loss (α) along an optical stripe are related to an intrinsic gain ($g_{i}$), i.e., $g=\Gamma g_{i}-\alpha$ [16].

While an optical stripe is made by a cylindrical lens, it is important to prepare a uniform beam stripe to avoid artifact. This can be realized by utilizing a beam expander and a rectangular slit, and the beam uniformity can be measured by image sensor or knife-edge method. Provided that the beam uniformity is given along the stripe, the initial photon density is constant before the amplification process begins, i.e., the same amplification process can be assumed at any infinitesimal segment (d*x*) along the stripe (Appendix A).

## 3. Results and Discussion

Figure 2a shows the ASE spectrum of Cd_0.6_Zn_0.4_Te/ZnTe DQDs with an 18 nm ZnTe separation layer as a function of increasing stripe length. In Figure 2b, stripe length dependence of the ASE intensity is plotted for three different emission energies, where two were selected from the PL peaks of LQDs (2.170 eV) and SQDs (2.254 eV), and the intermediate energy level (2.218 eV) was also chosen as a reference energy. As the stripe length increases, it is known that gain saturation appears gradually. Therefore, Equation (2) is valid within a short stripe length range, where gain saturation is negligible. Interestingly, we found the gain saturation depends on stripe length. Therefore, the fitting range can vary for different emission energies. In Figure 2c, the fitting function of Equation (2) shows an excellent agreement with the experimental data up to a 235 ± 35 μm stripe length.

For Cd_0.6_Zn_0.4_Te/ZnTe DQDs with a 6 nm ZnTe separation layer, the ASE spectra measured for increasing stripe length are shown in Figure 2c. For three selected emission energies, the ASE intensity was plotted for stripe length in Figure 2d, where one is selected from the ASE spectrum peak (2.191 eV) and two from the spectral wings (2.218 eV and 2.170 eV). Up to a stripe length of 300 ± 20 μm, we found Equation (2) fits very well. However, we found gain saturation occurs at relatively long stripe lengths compared to the result with the 18 nm ZnTe separation layer (235 ± 35 μm). This can be attributed to the density difference between LQDs and SQDs, where the density of LQDs (4 × 10^10^ cm^−2^) is twice the density of SQDs (2 × 10^10^ cm^−2^) [5]. With a large separation layer of 18 nm, the two different QDs become nearly independent. As a result, the ASE intensity ratio of LQDs to that of SQDs is roughly a factor of two (~2.6), as shown in Figure 2a.

On the other hand, with a small separation layer of 6 nm, tunneling becomes significant. In this case, LQDs and SQDs couple via tunneling, but some LQDs remain uncoupled due to the density imbalance. Therefore, the density of coupled QDs with the 6 nm separation layer is lower than the density of independent LQDs with the 18 nm separation layer. The density imbalance also affects gain saturation, i.e., as the number of QD emitters decreases along the stripe, gain saturation occurs at a relatively long stripe length.

In Figure 3a, we obtained optical modal gain spectra of Cd_0.6_Zn_0.4_Te/ZnTe DQDs with an 18 nm ZnTe separation layer, where the modal gain coefficients at 5 K were obtained by fitting with Equation (2). For example, given the three selected emission energies (2.254 eV, 2.218 eV, and 2.170 eV) in Figure 2b, we obtained corresponding gain coefficients of 125.2 ± 29.2 cm^−1^, 42.6 ± 33.6 cm^−1^, and 264.2 ± 15.4 cm^−1^, respectively. In the inset of Figure 3a, the energy levels and the wavefunctions of the ground state excitons are shown schematically for SQDs (blue) and LQDs (red), where the energy levels are determined separately regarding the width of the potential well, i.e., the lateral diameter of QDs, and the separation layer (~18 nm). The gain ratio of LQDs to that of SQDs is approximately twice, which is consistent with the density ratio between LQDs (4 × 10^10^ cm^−2^) and SQDs (2 × 10^10^ cm^−2^). Interestingly, we found that the optical gain at the intermediate energy (2.218 eV) is non-zero (42.6 ± 33.6 cm^−1^). This can be explained by size distribution and tunneling. In Figure 1b, the two PL spectra of LQDs and SQDs were separated without the emission at the intermediate energy, but the ASE appears at the intermediate energy. With weak excitation, the excitons of LQDs and SQDs seem to be barely coupled due to the suppressed tunneling arising from the large separation (~18 nm). However, as the excitation increases, state-filling effects become significant. This may activate a carrier transfer from SQDs to LQDs. This effect would be significant when the sizes of the SQDs and LQDs are similar.

Figure 3b shows the gain spectrum of Cd_0.6_Zn_0.4_Te/ZnTe DQDs with a small ZnTe separation layer (6 nm). The peak gain (123.9 ± 9.2 cm^−1^) appears at 2.191 eV, and the gain spectrum width is large compared to those of separate LQDs and SQDs with the large ZnTe separation layer (~18 nm) in Figure 3a. This is associated with the coupled energy level. When the separation between LQD and SQD is large enough, as shown in the inset of Figure 3a, it is very unlikely that the wavefunction tail of a QD extends across the barrier to the other QD. Therefore, each confinement level can be obtained separately, and the energy level is determined by the well width. As a result, the confinement energy difference between SQD and LQD is significant. On the other hand, when the barrier width is small, the two potential structures of LQD and SQD are inseparable due to the wavefunction overlap, and a single confinement level needs to be estimated regarding the asymmetric potential barriers. From a mathematical point of view, this can be realized through the diagonalization process, where the eigenstate is obtained from the initial states of the separate QDs. For intuitive understanding, two different constraints are involved. While the wavelength extension of SQD due to tunneling lowers the energy, the superposition of two states enforces an energy increase to the LQD for the bonding state.

With the small separation layer (6 nm), the tunneling becomes efficient. Therefore, gain appears at the coupled energy level, which is located in between the ground state levels of separate LQDs and SQDs. This result verifies that the coupled state can also give rise to optical gain. Recently, artificial quantum molecule engineering has been refined based on colloidal quantum dots [17], where the core–shell structure enhances the stability of an electron–hole pair, and the dot-to-dot separation can be controlled on an atomic scale. Since this is the case of strong confinement, various coupled states of excitons, trions, and biexcitons are likely formed across the small barrier width. In this case, those multi-exciton states may give rise to optical gain. Likewise, an assembly of quantum dots, the so-called supercrystal [18], can be associated with superradiance. In both cases, optical gain analysis is important as a prerequisite study.

When considering the size distribution of LQDs and SQDs, the gain spectrum width of the coupled QDs is expected to be similar to the ASE spectrum width. However, the gain spectrum for the small ZnTe separation layer (6 nm) was observed to be broad compared to the ASE spectrum width. The gain still appears at both low (<2.14 eV) and high (>2.22 eV) emission energy, and this can be attributed to uncoupled QDs. During the growth process, both LQDs and SQDs are located randomly. To be a coupled QD pair, the two QDs need to be aligned along the same vertical line. Otherwise, the distance between LQDs and SQDs increases, and they remain uncoupled. Additionally, note that the density of the coupled QDs is determined by the density of SQDs, and this restriction also affects gain. We found the peak gain (123.9 ± 9.2 cm^−1^) with a 6 nm separation layer is comparable to that (125.2 ± 29.2 cm^−1^) of the SQDs with an 18 nm separation layer. This result also supports the presence of uncoupled QDs.

## 4. Conclusions

Optical modal gain of Cd_0.6_Zn_0.4_Te/ZnTe DQDs with two different ZnTe separation layer thicknesses of 18 nm and 6 nm was investigated using a variable stripe length method. With a ZnTe separation layer of 18 nm, two dominant gain spectra from LQDs and SQDs were observed. However, with a separation layer thickness of 6 nm, a merged single gain spectrum appears as a result of the tunneling between LQDs and SQDs. Interestingly, we also found that gain saturation occurs at relatively long stripe lengths (~300 ± 20 μm) with a small separation layer thickness (~6 nm), and this can be attributed to the density imbalance between LQDs and SQDs. The peak gain (123.9 ± 9.2 cm^−1^) of the coupled QDs with a small separation (~6 nm) was also found to be comparable to that (125.2 ± 29 cm^−1^) of the SQDs in double QDs with a large separation of 18 nm. This result suggests the presence of uncoupled QDs in double QDs with a separation of 6 nm.

## Figures and Tables

**Figure 1 nanomaterials-13-00716-f001:**
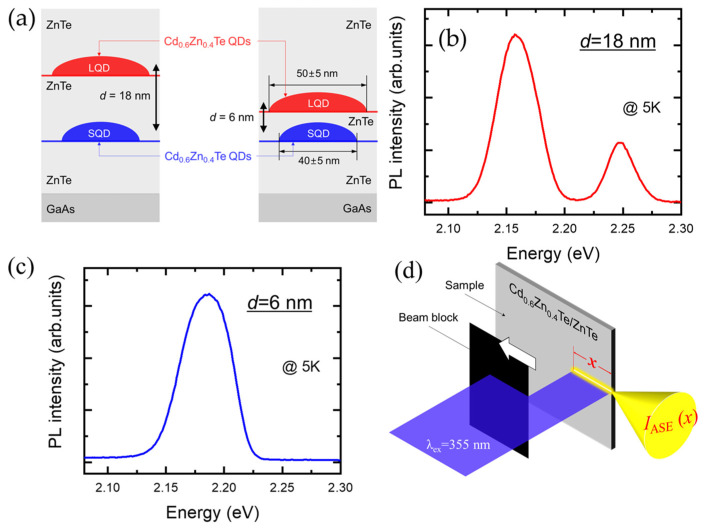
Schematics of DQDs, PL spectra at 5 K, and optical gain apparatus. (**a**) Structures of Cd_0.6_Zn_0.4_Te DQDs with two different ZnTe separation layer thicknesses of 18 nm (**left**) and 6 nm (**right**). PL spectrum of Cd_0.6_Zn_0.4_Te DQDs with a ZnTe separation of 18 nm (**b**) and 6 nm (**c**). (**d**) VSLM setup for optical gain measurement.

**Figure 2 nanomaterials-13-00716-f002:**
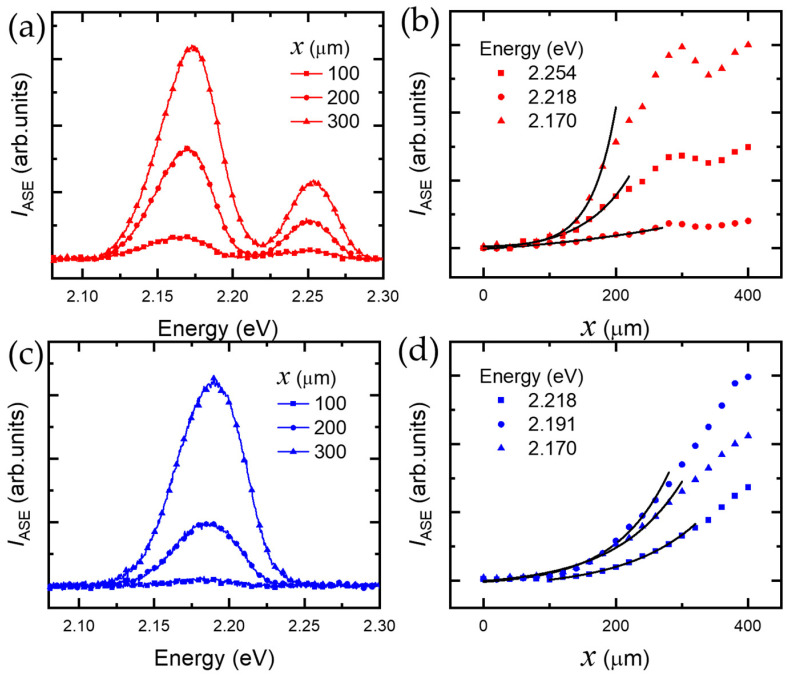
ASE spectra of Cd_0.6_Zn_0.4_Te DQDs at 5 K: (**a**) ASE spectra of Cd_0.6_Zn_0.4_Te DQDs with an 18 nm ZnTe separation layer for increasing stripe length (*x*), whereby the stripe length-dependent ASE intensity was obtained at various emission energies (**b**). (**c**) ASE spectra of Cd_0.6_Zn_0.4_Te DQDs with a 6 nm ZnTe separation layer for increasing stripe length, and stripe length-dependent ASE intensity at various emission energies (**d**).

**Figure 3 nanomaterials-13-00716-f003:**
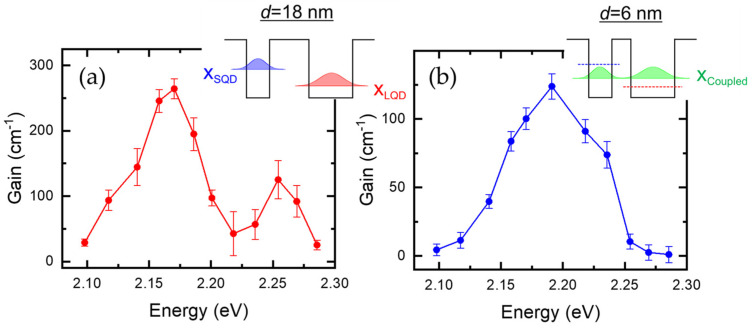
Optical modal gain of double QDs with different separation distances: Gain spectrum of Cd_0.6_Zn_0.4_Te DQDs with a ZnTe separation layer of 18 nm (**a**) and 6 nm (**b**), where each inset shows the exciton levels schematically.

## Data Availability

The data presented in this study are available on request from the corresponding author.

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
