# Peer review of "Optical Gain of Vertically Coupled Cd0.6Zn0.4Te/ZnTe Quantum Dots"

_nanomaterials, 2023, doi:10.3390/nano13040716_

Round 1

Reviewer 1 Report

1. In this manuscript, the author demonstrated that optical modal gain of Cd0.6Zn0.4Te/ZnTe double quantum dots was measured using a variable stripe length method, where large and small quantum dots are separated with a ZnTe layer. It is novel and interesting. However, the following questions should be addressed:

1. The grammar and writing skill should be improved.

2. In the schematic diagram of Figure 1d, the movement of beam block changes the size of the excitation beam, but also changes the excitation area and the size of the excitation gain. Can this process be simply understood as changing the excitation light energy density? How to understand their relationship and their impact on light glow?

3. What is the effect of X change on the coupling process between the two quantum dots? Why does the energy change be different?

4. Some latest references should be cited.

Reviewer 2 Report

This work describes Cd0.6Zn0.4Te/ZnTe double quantum dot's optical gain and underlying mechanisms involved with different separation distances from large and small QDs using variable stripe lengths. Indeed the work is well-presented, and the results are clearly presented. After a few minor corrections, I recommend this manuscript for publication in the Nanomaterials journal. 

I have a few comments to consider to improve further readability and clarity.

  1. Too many abbreviations (VSLM) can be written as it appears in 3 or 4 places throughout the manuscript. 
  2. Emphasize the vertical coupling details further in the discussion part. The current explanation is not clear. 
  3. Can you show an example microscopic image of ASE from any one stripe measured? This addition will certainly improve the quality. 
